# Ventilation Systems in Wetland Plant Species

Lars Olof Björn [1], Beth A. Middleton [2], Mateja Germ [3],* and Alenka Gaberščik [3]

1   Molecular Cell Biology, University of Lund, SE-223 62 Lund, Sweden; lars_olof.bjorn@biol.lu.se
2   U.S. Geological Survey, Wetland and Aquatic Research Center, 700 Cajundome Boulevard, Lafayette, LA 70506, USA; middletonb@usgs.gov
3   Biotechnical Faculty, University of Ljubljana, Jamnikarjeva 101, SI-1000 Ljubljana, Slovenia; alenka.gaberscik@bf.uni-lj.si
*   Correspondence: mateja.germ@bf.uni-lj.si

**Abstract:** Molecular oxygen and carbon dioxide may be limited for aquatic plants, but they have various mechanisms for acquiring these gases from the atmosphere, soil, or metabolic processes. The most common adaptations of aquatic plants involve various aerenchymatic structures, which occur in various organs, and enable the throughflow of gases. These gases can be transferred in emergent plants by molecular diffusion, pressurized gas flow, and Venturi-induced convection. In submerged species, the direct exchange of gases between submerged above-ground tissues and water occurs, as well as the transfer of gases via aerenchyma. Photosynthetic $O_2$ streams to the rhizosphere, while soil $CO_2$ streams towards leaves where it may be used for photosynthesis. In floating-leaved plants anchored in the anoxic sediment, two strategies have developed. In water lilies, air enters through the stomata of young leaves, and streams through channels towards rhizomes and roots, and back through older leaves, while in lotus, two-way flow in separate air canals in the petioles occurs. In *Nypa* Steck palm, aeration takes place via leaf bases with lenticels. Mangroves solve the problem of oxygen shortage with root structures such as pneumatophores, knee roots, and stilt roots. Some grasses have layers of air on hydrophobic leaf surfaces, which can improve the exchange of gases during submergence. Air spaces in wetland species also facilitate the release of greenhouse gases, with $CH_4$ and $N_2O$ released from anoxic soil, which has important implications for global warming.

**Keywords:** metabolic gases; greenhouse gases; aerenchyma; anoxic soil





## 1. Introduction

The aquatic environment holds special challenges for plant survival. The diffusion of gases in water is about $10^4$-fold slower than in air, so that aquatic plants must perform photosynthesis in water, and maintain aerobic respiration in flooded conditions [1,2]. Herbaceous wetland plants differ significantly, according to the accessibility of gases for their metabolism regarding their position in the water column. Researchers define various functional groups, namely, (1) emergent macrophytes or helophytes that are rooted in water-saturated soil, with foliage extending into the air (e.g., *Typha latifolia*, *Phragmites australis*); (2) floating-leaved macrophytes that are living in water rooted in hypoxic or anoxic sediment, with leaves floating on the water surface (e.g., *Nuphar luteum*, *Nymphaea alba*); (3) submerged macrophytes that grow completely submerged under the water, with roots or rhizoids attached to the substrate (e.g., *Myriophyllum spicatum*, *Potamogeton crispus*); and (4) free-floating macrophytes that float on or under the water surface, and are usually not rooted in the sediment (e.g., *Ceratophyllum demersum*) [3]. In addition, wetlands also host many different woody plants that are permanently or occasionally rooted in water-saturated sediment [4]. These species, belonging to different groups, often possess adaptations to overcome oxygen and carbon dioxide deficiencies, in order to maintain optimal conditions for photosynthesis and respiration. Emergent and floating-leaved species have an advantage over submerged species because their above-ground parts are fully

or partly exposed to air. Aerial leaves have stomata in their epidermis, which can be adjusted to optimize exposure of internal tissues to the atmosphere and the exchange of gases. Thus, aerial plant parts are well supplied with oxygen, but for roots and rhizomes anchored in water-saturated soils, oxygen for respiration can be limited. Therefore, efficient ventilation systems are crucial for their survival. Ventilation systems rely on a passive molecular diffusion process, on pressurized gas flow, or Venturi-induced convection [5]; however, in submerged plant tissue, the direct exchange of gases between these tissues and water also occurs [6]. In most aquatic species, ventilation is enabled by an extended system of air canals and intercellular spaces called aerenchyma, which develop in different plant organs from roots, to stems and leaves. [7,8]. Gases in aerenchyma can originate from the atmosphere, rhizosphere, or plant metabolism [9]. Laing [10] shows a strong relationship between the leaf area and the extent of changes of oxygen and carbon dioxide concentrations in aerenchyma during periods of illumination; thus, the contribution of metabolic gases may vary significantly among species.

Aquatic plants mainly form aerenchyma constitutively in different organs, namely, roots, leaves, and stems, while some amphibious and terrestrial plants produce aerenchyma in response to an oxygen shortage [7]. The presence of aerenchyma may differ among species. Independent of habitat, aerenchyma patterns are stable at the genus level, and the consistency of pattern is stronger in the roots than in the shoots [11]. In addition to the atmosphere, gases in aerenchyma can originate from the rhizosphere or plant metabolism [9]. The formation of aerenchyma may not depend on environmental conditions, or be induced by flooding [1]. Aerenchyma cells are formed lysigenously by programmed cell death, as is the case of rice roots; schizogenously by the expansion of intercellular spaces [11]; and expansigenously (secondary aerenchyma) by cell division or enlargement, without cell separation or death [12]. These enlarged spaces may develop either in primary tissues (primary aerenchyma), or in secondary tissues (secondary aerenchyma) [13]. According to Doležal et al. [14], lysigenous aerenchyma are mostly produced by submerged plants, schizogenous aerenchyma by terrestrial and perennial wetland plants, and expansigenous honeycomb aerenchyma by aquatic floating-leaved plants. The amount of intercellular spaces varies significantly among species. In aquatic species, these intercellular spaces contribute up to 60% of the leaf volume [15], while in mesophytes, their volumes range from 2–7% [16]. Thus, in non-tolerant species, flooding may result in the demise of the plant.

Beyond ventilation, aerenchyma cells have other important ecological functions, including acting to store gases and increasing their internal conductance to roots and shoots [7]. The transfer of oxygen to underground organs, via aerenchyma during soil flooding, may prevent the suffocation of plants. Oxygen can also be transferred from roots to the rhizosphere, via aerenchyma. This critically important oxygen to oxidize and detoxify toxic chemicals formed in sediments in environments with low redox potential [17,18], noting that a lack of oxygen is associated with reduced forms of sulfur, manganese, and iron that may reach toxic levels in the soil [6].

In wetland soils, gas concentrations of several gases, such as carbon dioxide and methane, exceed atmospheric concentrations. Thus, aerenchyma can also be a path for greenhouse gas emissions from the plant, as methane and nitrous dioxide are released via plants from waterlogged sediments to the atmosphere [19,20].

Some photosynthetic $O_2$ produced by submerged plants oxygenates the water column, while natant plants can prevent oxygen diffusion from the atmosphere to water [21–24]. Aerenchyma cells lend buoyancy and mechanical resistance to breakage, with a relatively small investment in biomass by aquatic plants [25].

The ventilation in wetland plants takes place via various plant structures, and is enabled by the presence of aerenchyma in these structures. The source of gases and influx and efflux locations may differ significantly among different species and plant groups. In this review, we summarize the outcomes of research, and point out the similarity, diversity,

and functional features of ventilation systems in various functional groups of wetland plants, and point out their potential effects on the wider environment.

## 2. Ventilation Mechanisms in Various Wetland Plant Groups

### 2.1. Submerged Species

In submerged aquatic species, the ventilation system is especially important since these have no direct connection with the atmosphere [26]. Extensive aerenchyma in the stems of all submerged plants that may be lysigenous and schizogenous (Figure 1) enable buoyancy for their flexible stems in water; however, this may limit the distribution of aquatic vascular plants to the depth where hydrostatic pressure does not compress stems [27]. Gases in submerged species come from plant metabolism, water, and sediment. The aerenchyma function as a reservoir for metabolic gases. Hartman and Brown [28], studying gas dynamics in *Elodea canadensis* Michx. and *Ceratophyllum demersum* L., detect a lag between peak values for dissolved oxygen in the surrounding water, and oxygen in the internal atmosphere. This situation is also true of the representatives of the genus *Lobelia* L., *Lilaeopsis* Greene, and *Vallisneria* L., which obtain more than 75% of their $CO_2$ needs from the sediment [29,30]. However, photosynthesis is important for internal $O_2$ status and aeration of anoxic sediments [31–33]. Sand-Jensen et al. [34] show that submerged macrophytes release oxygen from their roots during illumination, and at lower rates during darkness. The amounts of oxygen released varies among species from 0.04 to 3.12 μg $O_2$/mgDM/h; the species *Lobelia dortmanna* is the most effective. This involvement of photosynthesis enables aerobic metabolism of roots, sediment oxygenation due to radial oxygen loss, and improves nutrient uptake from the sediment [26].

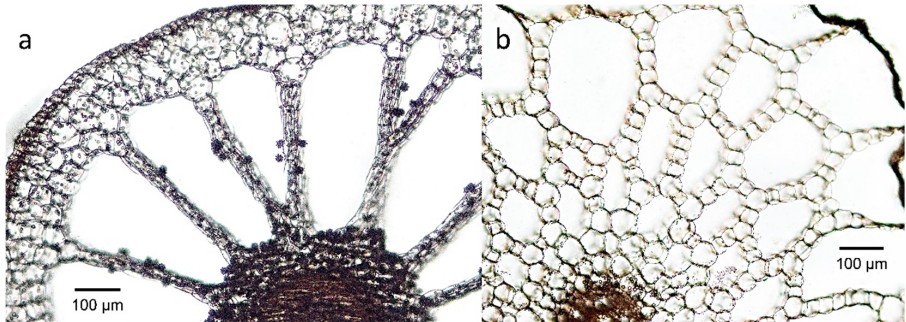

**Figure 1.** Transection of stems of two submerged species showing different types of aerenchyma; (**a**) lysigenous in *Myriophyllum spicatum* and (**b**) schizogenous in *Potamogeton crispus.* Photo: Matej Holcar.

A few outstanding representatives of submergent species include quillworts, *Isoëtes* L., which are the only surviving genus of a large plant group, distantly related to the clubmosses. This group emerged about 300 million years ago in ephemeral pools and oligotrophic lakes, and later radiated into terrestrial habitats [35]. Species in this group are well-adapted to cold environments, and limited light and carbon dioxide supply [36]. This genus represents the oldest lineage of plants with Crassulacean acid metabolism (CAM) [33].

At least some of species of the genus *Isoëtes* have air canals from the leaves through the periderm to the cortex aerenchyma in the stem [37]. Green [38] compares the air canals and aerenchyma in *Isoëtes* with the corresponding structures in Carboniferous–Permian arborescent lycopsids, such as *Lepidodendron* Stern. spp. The extinct lycopsid swamp trees called *Stigmaria* Stern. have similar air channels in their roots [39].

Most modern species of *Isoëtes* grow in wet environments, and use CAM to fix carbon. Due to the mineralization of organic matter, the $CO_2$ concentration in sediment is 10–250 times higher than atmospheric equilibrium [40]. Thus, these species uptake sedimentary $CO_2$ that is transferred via the large gas channels [38], which are present in various *Isoëtes* organs [41] (Figure 2). $CO_2$ leakage from gas spaces is prevented by the extremely high diffusional resistance of water. However, some *Isoëtes* species have an amphibious

character, and may develop leaves with stomata and switch to $C_3$ photosynthesis if plant parts emerge into the air from the water [35]. In addition, Pedersen et al. [42,43] show that various isoetids (small, aquatic plants with thick, stiff leaves or stems that form basal rosettes [42]), seagrasses, and rosette-bearing wetland species may have leaves buried in sediments, and their achlorophyllous bases may function as an exit for $O_2$ to the sediment, and entrance for $CO_2$ from the sediment [42,43]. This system may also affect the concentrations of gases in aerenchyma and photosynthesis [44].

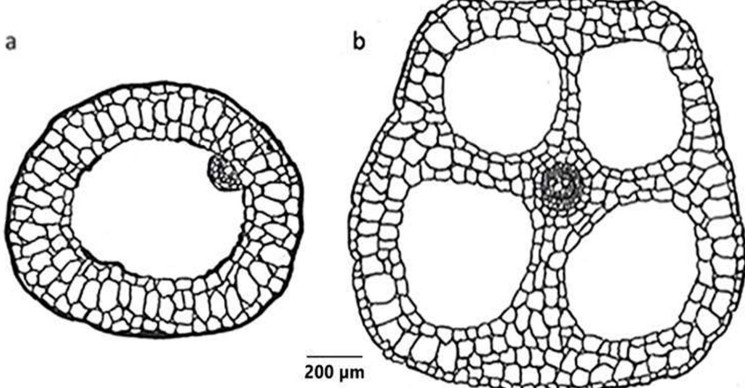

**Figure 2.** The root (**a**) and leaf (**b**) cross-section of *Isoëtes* L. sp. with large central air spaces. Drawings: Alenka Gaberščik.

### 2.2. Species with Natant Leaves

The advantage for these species is a direct connection with the atmosphere via natant leaves; however, since some of them grow in water as deep as 3 m [27,45], they need also effective aeration of the anoxic sediment.

Water lilies of the genus *Nymphaea* L. and *Nuphar* Sm. possess well-developed ventilation systems (Figure 3). Air enters through the stomata of young leaves that have just reached the water surface, streams through the channels of long petioles, through rhizomes and roots, and back to the external air through older leaves (Figure 4). This system accelerates $O_2$ flow from the atmosphere to the roots, and $CO_2$ and $CH_4$ flow from the roots to the atmosphere. This remarkable ventilation system has significant physiological and ecological consequences [46]. Laing [10] measures the daily dynamics of internal gas composition in various tissues of *Nuphar advena* (Aiton) W.T. Aiton, and finds a decrease in carbon dioxide and an increase in oxygen during the day, while the opposite is true at night. These findings show the important role of the metabolic process in influencing gas concentrations in various tissues.

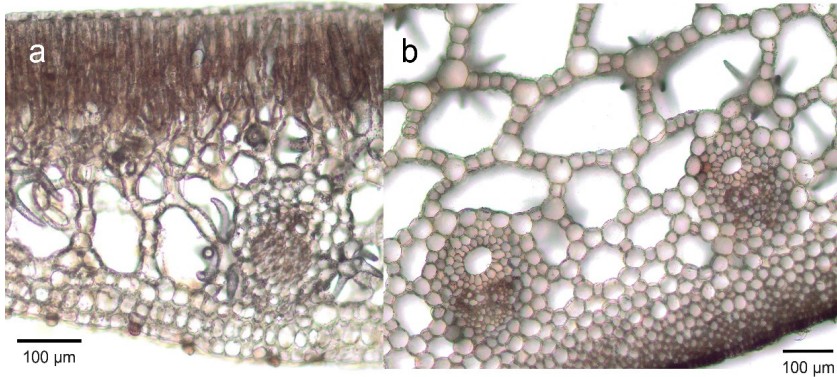

**Figure 3.** Transection of (**a**) natant leaf and (**b**) petiole of *Nuphar luteum*. Photo: Matej Holcar.

The ventilation is driven by the pressurization of the gas in air spaces of young natant leaves, due to thermo-osmosis of gases during cooling by transpiration [45,46]. Cooling

by transpiration of the upper leaf surface enables a steeper temperature gradient within the leaf, intensifying internal aeration [47]. The leaves are warmed by the sun during the day, and by water in the surrounding environment at night, which enables ventilation at night [48]. This process is also supported by specific morphological features that may differ among species [49]. One important feature is that gas transport through a small opening varies with the dimensions of the opening, depending on whether the gas transport takes place by diffusion or by mass flow. The rate of gas transport by diffusion is proportional to the diameter (or some other linear dimension if the opening or aperture is not circular), and to the difference in concentration of the gas on the two sides of the opening, with the slower movement of heavy vs. light gas molecules.

The rate of mass flow follows Poiseuille's law. The flow rate is proportional to the total pressure gradient between younger and older leaves [5], and the diameter (or corresponding measure) is raised to the fourth power. Therefore, diffusion often dominates in plant parts with small openings, and mass flow when the openings are large [46]. The openings in young leaves are narrow. According to Grosse and Schröder [50], the important openings, in this case, are not the stomata ($\approx 5.6 \ \mu m \times 2.4 \ \mu m$), but narrow passages (intercellulars) ($\approx 1 \ \mu m$) between cells inside the leaves. The oxygen concentration outside the leaves is higher than in the airspaces inside the leaves. This situation is partly because the air inside is diluted by water vapor, and is almost saturated in the intercellular spaces. Therefore, oxygen diffuses into the young leaves, and the total gas pressure there exceeds that of the external air. As the passages to the higher-oxygen external air are so narrow, little air escapes from the young leaves to the external atmosphere. It is easier for air to flow through the wider "pipes" in the petioles down to the rhizomes and roots, where oxygen is consumed by respiration. The rest of the air streams out to the atmosphere through the older leaves, where the passages are wider than in the younger leaves.

The streaming of air in the yellow water lily, *Nuphar luteum* (L.) Sm. is described not only by Schröder et al. [51], but also in a series of articles by John Dacey and coworkers, and the results are summarized by Dacey [46]. Richards et al. [45] perform similar studies on *Nymphaea odorata* Aiton. In the case of *N. luteum,* the network of internal gas spaces enables a pressurized flow-through system, that directs air rich with oxygen from young leaves to the underground organs, and with air rich in carbon dioxide back via the older emergent leaves to the atmosphere [46]. Konnerup et al. [52] compare pressure differences and convective gas flow in twenty species of tropical angiosperms (Table 1). The highest flow rates are found for *Nymphaea rubra* Roxb. and *Nelumbo nucifera* Gaertn.. Some species achieving large pressure differences have low flow rates. The two *Eleocharis* R. Br. species differ remarkably with respect to the pressure differences produced.

**Table 1.** Comparison of pressure differences and convective gas flow in twenty species of tropical angiosperms (adapted from Konnerup et al. [52]).

| Plant | ΔP(Pa) | Air Flow (mL/min) | Plant | ΔP(Pa) | Air Flow (mL/min) |
|---|---|---|---|---|---|
| **Monocotyledons** | | | **Dicotyledons** | | |
| *Cyperaceae* | | | *Nymphaeaceae* | | |
| *Cyperus compactus* Retz. | 20 | 0.60 | *Nymphaea rubra* Roxb. | 236 | 140 |
| *Cyperus digitatus* Roxb. | 14 | 1.29 | *Nymphaea nouchali* Burm. f. | 116 | 15.2 |
| *Eleocharis dulcis* (Burm. f.) Trin. ex Hensch | 628 | 11.9 | *Nelumbonaceae* | | |
| *Eleocharis acutangula* Schultes | 15 | 0.10 | *Nelumbo nucifera* Gaertn. | 295 | 288 |
| *Scirpus grossus* L. f. | 3 | 0.22 | *Menyanthaceae* | | |
| *Scirpus littoralis* Shrad. | 83 | 0.39 | *Nymphoides indica* (L.) Kuntze | 485 | 36 |
| *Scleria poaeformis* Retz. | 22 | 1.11 | *Convolvulaceae* | | |
| *Poaceae* | | | *Ipomoea aquatica* Forssk. | 3 | 0.18 |
| *Phragmites vallatoria* (L.) Veldkamp | 482 | 1.59 | | | |
| *Urochloa mutica* (Forsk T.Q. Nguyen | 11 | 0.09 | | | |
| *Hymenachne acutigluma* (Steud.) Gilliland | 141 | 0.55 | | | |

**Table 1.** *Cont.*

| Plant | ΔP(Pa) | Air Flow (mL/min) | Plant | ΔP(Pa) | Air Flow (mL/min) |
|---|---|---|---|---|---|
| **Monocotyledons** | | | **Dicotyledons** | | |
| *Oryza rufipogon* Griff. | 23 | 0.32 | | | |
| *Leersia hexachloa* Swartz. | 62 | 0.15 | | | |
| Pontederiaceae | | | | | |
| *Eichhornia crassipes* (Mart.) Solms | 8 | 0.12 | | | |
| Araceae | | | | | |
| *Colocasia esculenta* (L.) Schott | 3 | 0.10 | | | i |
| Limnocharitaceae | | | | | |
| *Limnocharis flava (L.)* Buchenau | 6 | 0.81 | | | |

Lotus (*N. nucifera*) is not closely related to water lilies, but has a similar growth form, and ventilates its organs in oxygen-deficient sediment by streaming air. The thermo-osmotic gas transport depends on a temperature differential between the air in the aerenchyma of the leaves and the surrounding atmosphere [53]. Stomata located in the central parts of leaves have an important function in the active regulation of airflow [54]. These central stomata are three times larger than stomata of the other part of the leaf lamina, and control the gas release by opening and closing [55]. Lotus differs from other groups because each petiole has separate channels for descending and ascending air [53,56]. Air is, thus, transferred from leaves, down through petioles, rhizomes, and then back to the atmosphere through large stomata [54] (Figure 4).

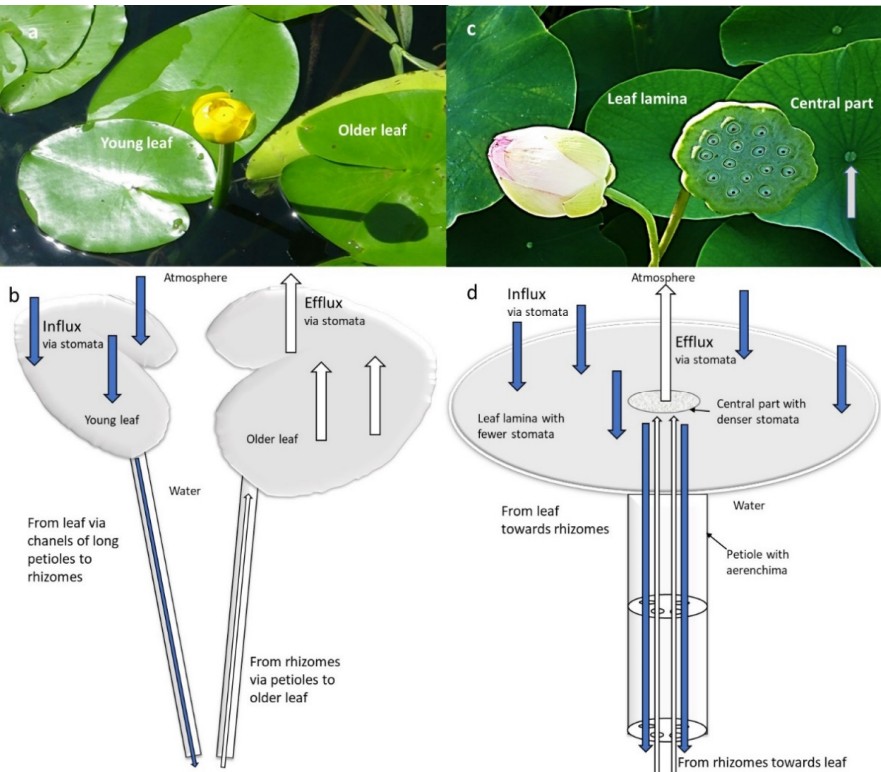

**Figure 4.** Ventilation system in *Nuphar luteum* (**a**,**b**) *Nelumbo nucifera* (**b**,**c**). (**a**) *N. luteum* young and older leaves; (**b**) the scheme of ventilation system where arrows shows that air from atmosphere enters via young leaves through stems with aerenchyma to rhizomes, and via older leaves back to the atmosphere; (**c**) *N. nucifera* leaves with visible central part with denser stomata; (**d**) the scheme of ventilation system with arrows indicating the direction of flow (blue—influx from atmosphere via stomata in leaf lamina through aerenchyma in leaf and petiole to aerenchyma in rhizomes; light grey—efflux from rhizomes through aerenchyma in petiole to the central part of the lamina, and finally to atmosphere). Photos and drawings: Alenka Gaberščik.

*Polygonum amphibium* L. is an amphibious plant with natant leaves. and thrives in water up to 2 m, with its hollow stems that enable the aeration of underground organs (Figure 5). However, gases trapped in the aerenchyma may also benefit the photosynthesis and lower photorespiration rate [57], due to changes in the ratio of internal $CO_2/O_2$ concentrations.

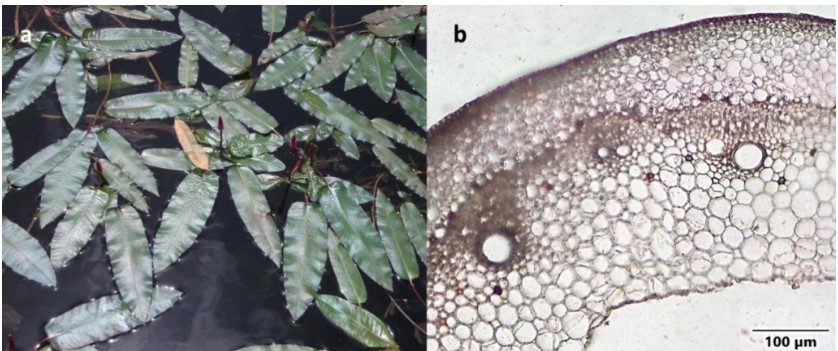

**Figure 5.** *Polygonum amphibium* L. f. natans; (**a**) natant leaves and (**b**) section of a hollow stem. Photo: Matej Holcar.

### 2.3. Helophytes

Helophytes live anchored in water-saturated sediment, while their above-ground organs are usually in air. Most of them have well-developed aerenchyma [1]. When leaves of different helophytes are submerged, they produce gas films at the leaf surface to improve photosynthesis. Colmer and Pedersen [58] show that *Phalaris arundinacea*, *Phragmites australis*, and *Typha latifolia* form gas films on both leaf sides, *Glyceria maxima* form gas films on the adaxial only, and *Acorus calamus* and *S. emersum* do not form gas films.

Rhizomes and stems of wetland representatives of horsetail (*Equisetum*) have large canals [59]. Internal ventilation systems differ among *Equisetum* species, and are best developed in the great horsetail, *E. telmateia* Ehrh. In this species, an airflow of 120 mL/min and a wind speed of 10 cm/s were measured [60], and this flow may be evaporation-driven. Air enters via stomata on branches, passes to substomatal cavities, and then via intercellulars to aerenchyma channels in branches, to the main stem, rhizome, and cortical intercellular spaces of roots [61].

Methane release from *E. fluviatile* L. stands suggests that this species lacks a pressurized ventilation system because no diurnal changes are detected [59]. In addition, no or very low airflow is measured in this species, even though it thrives in deep water [62]. The power for the airflow comes from the evaporation of water inside the plant. If the air surrounding the plant is saturated with water vapor, the flow stops, and it is sped up by wind that evaporates water vapor from the plant surface [61]. For example, *E. palustre* L. maintains an inner mass flow of up to 13 mL/min (Figure 6). No flow is found in *E. sylvaticum* L. or *E. arvense* L., perhaps because these species lack air channels in the branches [60]. These species only rely on diffusion for oxygen provision to their below-ground parts, which is facilitated in environments of well-aerated soil.

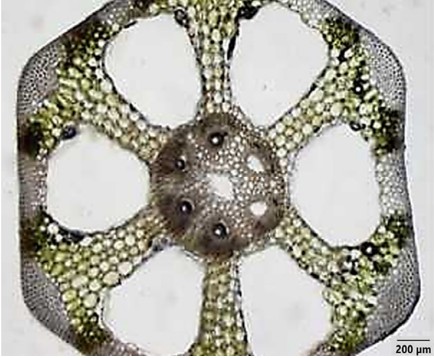

**Figure 6.** The cross-section of marsh horsetail (*Equisetum palustre* L.) stem. Photo Matej Holcar.

Common reed and other *Phragmites* species frequently grow in water-saturated anoxic soil. *Phragmites* are ventilated by "compressed air" in a similar way to water lilies and lotus [63,64], but additional aeration may be obtained by suction from old broken reeds when the wind is blowing over them, a Venturi-effect [65], and also by oxygen from photosynthesis. The oxygen pressure in the rhizomes rises rapidly in the morning, peaks near midday, and declines slowly to a minimum at 6 am [66]. Three factors probably contribute to the midday peak including light (photosynthesis), temperature, and the difference in water vapor pressure between the external air and the intercellulars. External humidity is at its lowest in the early afternoon, when humidity in leaf intercellulars (due to insolation) is highest. Oxygen from roots is released in the nearby rhizosphere, which is known as radial oxygen loss [67], and oxidizes different toxic substances produced in anoxic soils that can harm soil biocenosis. This situation is also the case in *Phragmites australis* (Cav.) Trin. ex Steud, where the aeration system increases rhizosphere oxygenation and lowers toxic sulfide concentrations [68]. Some species prevent excessive oxygen loss from roots by forming a barrier in their epidermis, exodermis, or subepidermal layers [69].

Konnerup et al. [52] also find convective ventilation in *P. vallatoria* (L.) Veldkamp, and eight other wetland plant species. The only other monocotyledon species for which a similar pressure difference is found is *Eleocharis dulcis* (Burm. f.) Trin. ex Hensch (*Cyperaceae*), but the flow rate is lower.

In *Scirpus lacustris* (L.), sediment-derived $CO_2$ in aerenchyma (Figure 7) presents an important source of inorganic carbon used for photosynthesis in submerged green stems, especially before they reach the water surface [70]. In this species, leaves are usually reduced, however, leaf blades up to 100 cm long can develop under water [71].

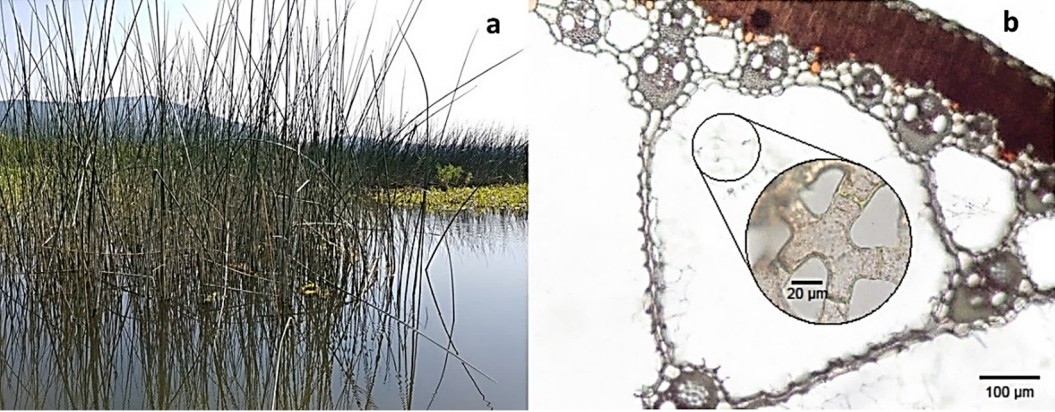

**Figure 7.** (**a**) Leafless *Schoenus lacustris* stems; (**b**) transection of stem with lacunae filled with thin-wall aerenchymal parenchyma. The magnification of parenchymatic cells is shown in the magnified circular insert. Photo: Matej Holcar.

In plants with reduced pressurized ventilation, the growth rate is diminished, possibly due to reduced mineral uptake and availability, a poorly developed root system, or impaired root function [63]. Studies in the USA reveal differences in function between native and non-native *P. australis* types. Ventilation efficiency is 300% higher per unit area for non-native types in comparison to native types, due to the increased oxidation of the rhizosphere [72].

The cattails (*Typha* L., Figure 8) are phylogenetically related to reeds. *Typhaceae* and reeds ventilate their below-ground parts in similar ways. An air stream of 8 mL/min per leaf is observed for *T. latifolia*, and 3.5 mL/min for *T. angustifolia* L. [73]; however, some leaves can function with influx or efflux in changing environments. A flux of 11 mL/min is observed, compared to a maximum of 60 mL/min through a water lily petiole [46]. Contrary to the situation in lotus, for *Typha*, the stomata constitute the pressure-generating pores [74]. The stomata are most effective for creating an inner pressure when they are partially closed, and provide almost no pressure when they are open to 0.3 μm wide. Air enters through middle-aged leaves, and exits through the oldest ones [73].

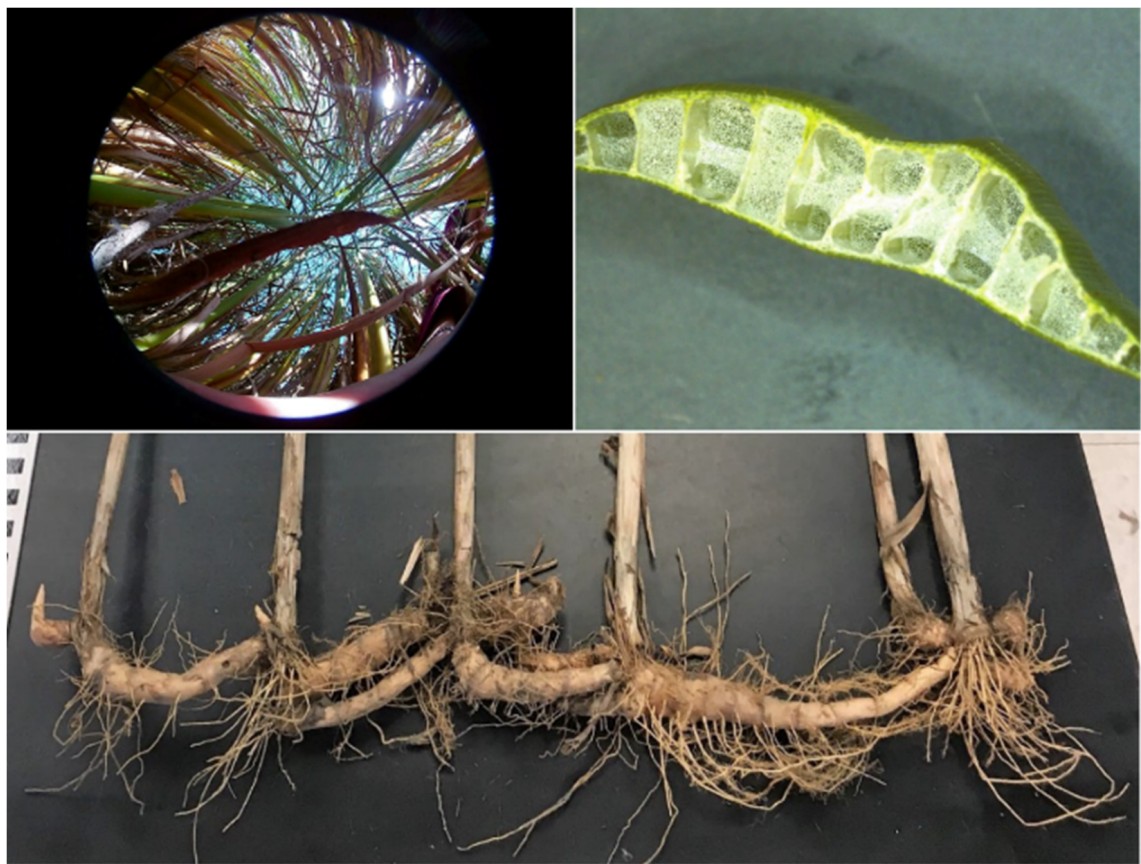

**Figure 8.** Cattail, *Typha* L. sp. Top left, shoots seen from below. Top right, a cross-section of a leaf with large air spaces, which are connected to air spaces throughout the plant. Bottom, rhizomes. Reproduced with permission from Bansal et al. [75].

Duarte et al. [76] claim that oxygen in the root area is not obtained exclusively from the atmosphere or photosynthesis in the leaves, but, in addition, oxygen in the root is supplemented by the decomposition of hydrogen peroxide by catalase. The study of Laing [10] reveals that the shading of *T. latifolia* has little effect on gas composition in air spaces of the submerged stolon; however, a slight decrease in oxygen concentration is observed in leaves. In hypoxic growth conditions, various *Typha* species behave in diverse ways; *T. angustifolia* increases its root porosity and root mass ratio, while *T. latifolia* increases its root diameter [77].

Rice (*Oryza sativa* L.) can grow in deep water ([78]). Under complete submergence, rice leaves and stems elongate significantly to reach the air–water interface; however, this may exhaust its energy reserves, and cause death if deep water persists for a long time [79]. Rice mitigates waterlogging stress by forming lysigenous aerenchyma and a barrier to prevent radial $O_2$ loss from roots, in order to supply $O_2$ to the root tip [80]. However, it differs from the species described above in the transport of air. Air is partly transported through a layer of air on the surface of the leaves. At least in the majority of rice cultivars, the leaves are water-repellent on both sides. A layer of air is formed on these leaves, up to 25 μm thick [81], and with a volume that is 44% of that of the leaf (Figure 9). Gas molecules in the thin layer are transported forward by diffusion at a greater pace than if they could move in all directions, but there may also be some mass flow. A layer of air at the leaf surface also contributes to the tolerance against submergence in the case of *Spartina anglica* C.E. Hubb. [44]. The floating leaves of the grass *Glyceria fluitans* (L.) R. Br. have a lower cuticle resistance for dark $O_2$ uptake on the wettable abaxial side, compared to the hydrophobic adaxial side [82].

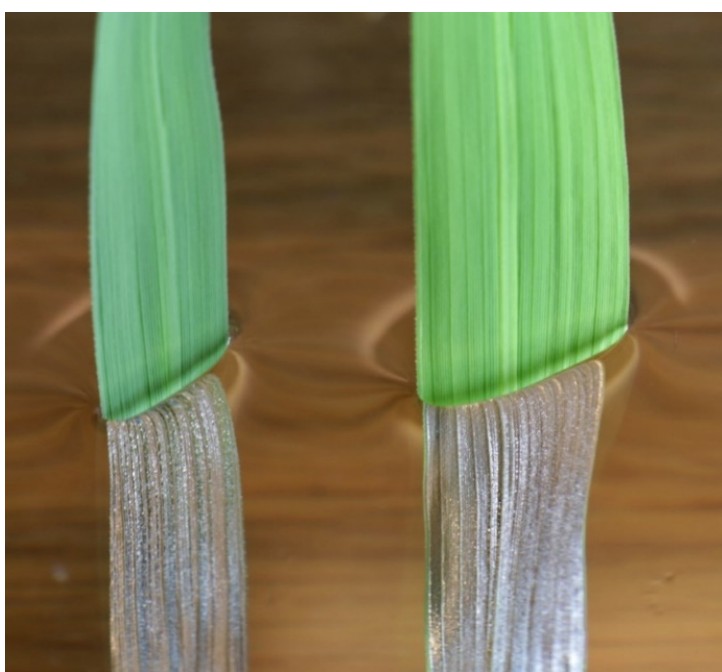

**Figure 9.** Rice (*Oryza longistaminata* A. Chev. & Roehr.) leaves protrude through the water surface, covered below the water surface by a layer of air. Photo by Ole Pedersen, University of Copenhagen. Cf. the cover of Science Vol. 228, No. 4697, 19 April 1985.

Rice also has large air spaces inside both leaves and roots, while the transition between shoot and root provides greater resistance to flow [83]. In rice, aerenchyma forms constitutively under aerobic conditions, while their formation is further induced under oxygen-deficient conditions [84]. Aerenchyma formation is promoted by ethylene production in adventitious roots [8] and in internodes, and this aerenchyma formation is mediated by reactive oxygen species (ROS) [85]. Little is known about the signaling and molecular regulatory mechanisms of ROS during plant-programmed cell death [86]. In the case of the fast-elongating 'Arborio Precoce' variety, ethylene controls aerenchyma formation, while in the variety 'FR13A', ROS accumulation plays an important role [87]. The study shows that the SNORKEL1 and SNORKEL2 genes that promote internode elongation are exclusively present in the genomes of deep-water rice [88].

Under $O_2$ deficiency, the porosity of adventitious roots enhances $O_2$ diffusion towards the root tip [89]. Flow takes place because the respiring plant replaces oxygen with carbon dioxide, which at pH = 7 and 25 °C has a solubility in water that is 140 times that of oxygen [78]. The study of Colmer and Pedersen [90] reveals that the dynamics in $pO_2$ within shoots and roots of submerged plants are related to periods of darkness and light, which supports the connection of underwater photosynthesis and internal aeration.

In *Pontederia cordata* L., the innermost layer of ground meristem in adventitious roots forms the endodermis and aerenchymatous cortex [91]. The comparison of $pO_2$ in submerged roots of *Rumex palustris* Sm. (flood-tolerant) and *R. acetosa* L. (flood-intolerant) reveals that roots of the former remain oxic, while in the latter, root $pO_2$ drops significantly (to 4.6 and 0.8 kPa, respectively) [92].

*2.4. Mangrove Forest*

Ferns of the genus *Acrostichum* L. (closely related to *Ceratopteris*) live rooted in mangrove vegetation and other settings [93]. All parts of these plants have large air spaces, including the roots. Aerenchyma are found in the roots of *Acrostichum* fossils, which suggests a coastal paleoenvironment [94]. Fonini et al. [95] also report large aerenchyma in the leaves of species *A. danaeifolium* Langsd. & Fisch., which is a feature endemic to mangrove.

A species of palm, *Nypa fruticans* Wurmb. or "snorkeling palm" adapted to life in mangrove forests; usually only the leaves appear above the water surface. These leaves eventually abscise when aging, but the leaf bases remain, and function as air inlets by developing a network of lenticels covering the leaf base [96]. These lenticels are connected with expansigenous aerenchyma [96]. The lenticels allow gases through, but prevent the entry of fluid water. The air spaces of the lenticels are connected to air spaces via the underwater stem to the roots. The first- and second-order roots have a small central stele surrounded by a wide zone of cortex with very spacious air channels, which develop after the separation and dissolution of cells. The mature crown may contain 6 to 8 living leaves, and 12 to 15 bulbous leaf bases at a time [97]. These leaf bases can function for up to 4 years after leaf abscission [96].

The main constituent of mangrove forests includes various types of mangrove species, which are not all closely related. Various mangrove species developed a variety of unique adaptations [98]. Most species possess structures, such as pneumatophores, knee roots, or stilt roots, which provide ventilation during low tides [99]. Studies by Scholander et al. [100] reveal that high oxygen pressure in the roots of *Rhizophora* L. is maintained via ventilation through the lenticels of the stilt roots. The cortex of *Rhizophora* roots have interconnected schizogenous intercellular spaces, extending to within 150 μm of the tip [101].

Pneumatophores act as "snorkels" from the roots of *Sonneratia apetala* Banks (Figure 10). Pneumatophores have numerous lenticels connected with aerenchyma; however, the pneumatophore development phase significantly affects the volume of gas spaces, since young pneumatophores have a less-developed lacunose cortex, in comparison to mature pneumatophores [98]. In *S. alba* Sm., root porosity in different root types ranges from 0 to 60%. Cable roots and pneumatophores have the highest ratio of aerenchyma per area (50–60%) [101]. The changes in cortex cells with developing gas spaces suggests schizogenous and lyzogenous formation of aerenchyma [102].

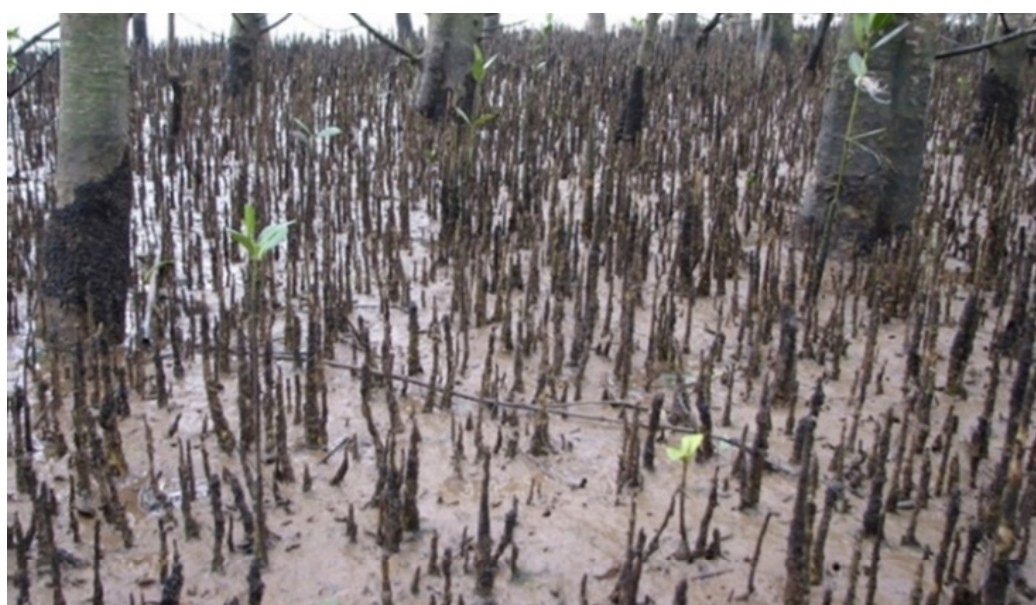

**Figure 10.** Pneumatophores of the mangrove tree *Sonneratia apetala* Buch. Ham. From Zhang et al. [103]. Creative Attribution license.

Comparing mangrove species belonging to various orders, Cheng et al. [104,105] find that the more air spaces the roots possesses, the greater the flooding tolerance. Curran et al. [106] conclude that *Avicennia marina* (Forsk.) Vierh. does not need airflow because the diffusion of oxygen satisfies the oxygen requirement. Oxygen enters the pneumatophores of this species through lenticels and "horizontal structures" on the pneumatophores [107]. The volume of gas spaces determined in the roots of this species is 40–50% [108].

### 2.5. Other Wetland Species

Baldcypress (*Taxodium distichum* (L.) Rich) is a conifer in the south–east of the USA, which can grow under both very wet and dry conditions. Growing in water, *T. distichum* develops "knees" emerging from the roots, which may be involved in oxygen transfer (Figure 11) and carbohydrate storage [109]. Under root and knee submergence, internal $O_2$ concentrations are significantly higher than during drawdown [110]. The knees do not possess gas conduits, so Rogers [109] concludes that knees serve to aerate the phloem in the inner bark for the oxygen requirements of the phloem, and downward conduction of oxygen dissolved in the phloem sap. Wang and Cao [111] report that flooding in *T. distichum* increases the porosity of the roots, stems, and leaves and, consequently, enhances $O_2$ diffusion to roots.

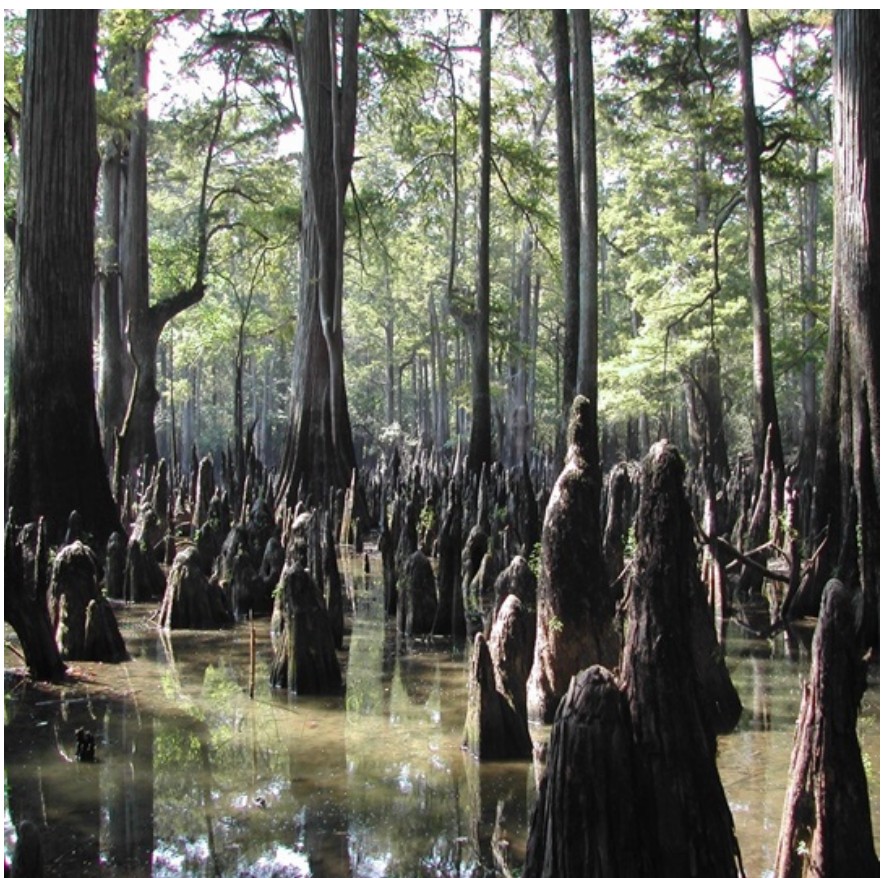

**Figure 11.** "Knees" emerging from the roots of baldcypress, *Taxodium distichum* (L.) Rich. in Arkansas, USA. From Middleton [112]. CC BY 4.0 license.

Some tree species in wetlands develop adventitious roots from the trunk during the rainy season, as an adaptation to the flooded environment. In *Syzygium kunstleri* (King) Bahadur and R.C. Gaur that is native to Borneo and Malaya, oxygen transportation occurs through aerenchyma in the root tips, periderm near the root base, and secondary aerenchyma between layers of phellem [113].

The roots of species colonizing flood forests and riparian zones are often subjected to water-saturated soil. Schröder [51] and Grosse and Schröder [4,50] find a thermo-osmotically driven gas flow in *Alnus glutinosa* (L.) Gaertn. from the external air, through the stems to the roots. A temperature difference of up to 3.6 degrees exists in other *Alnus* Mill. species between the intercellular spaces and the external air [114]. As the temperature difference increases from 0.6 to 3.6 °C, due to increasing light (from 50 to 200 µmol m$^{-2}$ s$^{-1}$), the pressure difference between the external air and the intercellular spaces in the stem increases from 5 to 17 Pa. An efficient thermo-osmosis process needs a narrow passage,

comparable in width to the mean free path lengths of the gas molecules (ca. 0.1 μm), between two compartments of different temperatures. However, some pressurization is possible with larger pores [115]. Details of thermo-osmosis of gases are provided by Denbigh [116], and Denbigh and Raumann [117]. In short, the pressures p1 and p2, which can develop in two compartments (designated 1 and 2), are determined by the equation:

$$\ln\left(\frac{p1}{p2}\right) = K*\left(\frac{1}{T1} - \frac{1}{T2}\right)$$

where T1 and T2 are the absolute temperatures of the two compartments, and K a constant.

The constrictions in the pathway that are needed for the development of thermo-osmotic gas flow in alder are related to the lenticels in the bark of the stem [118,119]. Gas flows from the colder external atmosphere to the warmer intercellular spaces.

In contrast, Armstrong and Armstrong [120] use *A. glutinosa* as an experimental species, and find that the thermo-osmotic gas flow is incompatible with the species plant anatomy, and so claim that roots are supplied with oxygen produced by chlorophyll-containing stems. The apparent discrepancy in the results of Armstrong and Armstrong vs. Grosse's group may be due to a difference in the age of the plants used by the two research groups, or to differences in submergence times. But the experiments by Dittert et al. [119] offer another explanation. They find that under experimental conditions, when the temperature of the trunk is kept above the temperature of the air, thermo-osmotic gas transport takes place, while under natural conditions (in Germany), there is only transport by diffusion, at all times of the year. Batzli and Dawson [121] find that in *Alnus rubra* Bong., lenticels develop (in this case on roots and root nodules) after 50 days of flooding.

### 3. Diversity of Ventilation Systems

Similar physical processes of ventilation occur in different taxonomical and functional plant groups that thrive in oxygen-deficient sediment; however, their morphological adaptations differ significantly (Table 2). Functional traits of these plants are not only the species' response to specific environmental conditions, but they also depend on their phylogeny. Jung et al. [11] find specific trends of aerenchyma patterns in several taxa of higher plants, and show that these patterns partially coincide with their phylogeny. The study of Cape reeds reveals that the presence of aerenchyma correlates with the eco-hydrological niche at the population and species level, indicating that waterlogging presents an environmental filter that excludes species without aerenchyma [122]. Bedoya and Madrinán [123], studying the evolution of the aquatic habit in genus *Ludwigia* L., find a convergence towards the absence of secondary growth in roots, smaller proportion of lignified tissue area in underground organs, and the presence of primary aerenchyma. However, there are also studies that are not consistent with these results. For example, the study of different *Carex* L. species in a phylogenetic context, with an even sampling across the different clades, shows that the size of the aerenchyma has only a weak relation to soil moisture [124]. *Carex* species with poorly developed aerenchyma have low performance in flooded soil, while partial submergence may even affect species with a larger amount of root aerenchyma [125].

**Table 2.** Ventilation mechanisms in different taxonomical and functional plant groups that thrive in oxygen-deficient sediment.

| Plant Group | Taxonomic Group | Source of Gasses | Ventilation Principle | Special Features | Reference |
|---|---|---|---|---|---|
| Submerged | Isoetids | Water, metabolism, Sediment | Diffusion, aeration of rhizosphere via buried leaves | Aerenchyma, CAM | [38,44] |
| | Angiosperms | Water, metabolism, Sediment | Diffusion | Metabolic gasses trapped in aerenchyma | [29,31] |

**Table 2.** *Cont.*

| Plant Group | Taxonomic Group | Source of Gasses | Ventilation Principle | Special Features | Reference |
|---|---|---|---|---|---|
| Floating | *Nuphar* spp., *Nymphaea* spp. | Air, metabolism, Sediment | Pressurized ventilation, thermo-osmotic gas transport, | 'Heat pump' drives gasses from the atmosphere via young natant leaves, petioles to roots and back, via older leaves to the atmosphere | e.g., [45,48,51] |
| | *Nelumbo nucifera* | Air, metabolism, Sediment | Pressurized ventilation, influx via laminal stomata of natant leaves through aerenchyma to rhizomes; back from rhizomes through aerenchyma in petiole through stomata in leaf central part | Leaf lamina with fewer and smaller stomata, leaf central part with larger and denser stomata, which actively regulate the airflow by opening and closing | [53–55] |
| Helophytes | *Equisetum* spp.—4 out of 9 have through-flow convection | Air, possibly also metabolism, sediment | Pressurized ventilation, humidity-induced diffusion, | Air moves through stomata through branches, via interconnecting aerenchyma channels in stem and rhizomes, with venting through the previous year's stubble or damaged shoot. | [60,61] |
| | *Phragmites* spp. | Air, possibly also metabolism, sediment | Pressurized ventilation, suction via old broken stems (Venturi-effect), air films on leaves when submerged | Via leaves, stems to root system, partly to sediment (ROL), and back to stems, leaves, and atmosphere | [58,63–65,126] |
| | *Typha* spp. | Air, sediment, possibly metabolism, oxygen in the rhizosphere may be obtained from the decomposition of hydrogen peroxide by catalase | Pressurized ventilation, leaf stomata create inner pressure, air films on leaves when submerged | Air enters through middle-aged leaves, and exits through the oldest ones | [58,73,74,76] |
| | *Oryza sativa* | Air films on leaves when submerged | Flow from above-ground parts via roots by diffusion, and possibly also by mass flow | Water-repellent leaf surface; air layer up to 25 μm, large air spaces inside leaves and roots, the porosity of adventitious roots, a barrier in roots to prevent radial $O_2$ loss from roots | [81,83,89] |
| Species of mangrove forest | *Acrostichum* spp. | | | All plant parts have large air spaces | [94,95] |
| | *Nypa fruticans* | | Bases of abscised leaves function as air inlets, by developing a network of lenticels covering the leaf base connected to aerenchyma | "snorkeling palm" leaf bases function up to 4 years after leaf abscission | [96] |
| | Mangroves | | High oxygen pressure in the roots is maintained via ventilation through the lenticels on different root structures connected with aerenchyma | Special structures, i.e., pneumatophores, knee roots, stilt roots, or plant roots, provide ventilation during low tides | [99] |

**Table 2.** *Cont.*

| Plant Group | Taxonomic Group | Source of Gasses | Ventilation Principle | Special Features | Reference |
|---|---|---|---|---|---|
| Other wetland species | *Alnus* spp. | Thermo-osmotically driven gas flow | In *Alnus glutinosa,* the flow is from the external atmosphere through the stems to the roots | Thermo-osmotic flow in alder is related to the lenticels in the bark of the stem, stem photosynthesis | [4,51,120,127] |
| | *Taxodium distichum* | | "knees" emerging from the roots to the surface of the water, flooding increases the porosity of roots, stems, and leaves, and enhanced $O_2$ diffusion to roots. | Snorkeling | [109] |
| | *Syzygium kunstleri* | | Oxygen transportation occurs through aerenchyma in the root tips, periderm near the root base, and secondary aerenchyma between layers of phellem | | [113] |

Most species use aerenchyma as a ventilation path, and as a reservoir for gases originating from the atmosphere, soil, and metabolic processes (e.g., respiration, photorespiration, and photosynthesis). The differences among groups are related either to specific environmental conditions (e.g., emergence, submergence), to specific species anatomy, as is the case for the lotus, or to metabolic processes and properties related to their growth form (e.g., emergent, submerged, floating-leaved, rosette, woody plants).

## 4. Plant Role in Emission of Greenhouse Gases

Ventilation systems are also important in the emission of greenhouse gases. We concentrate on methane ($CH_4$), although nitrous oxide ($N_2O$) is also important. Although the global warming potential (GWP) of methane is much greater than that of nitrous oxide over brief time intervals, nitrous oxide has about twice the warming potential of methane at a time horizon of 100 years (IPPC 2013), because it persists longer in the atmosphere. This section focuses on the role of plants rooted in water-saturated soils in the emission of greenhouse gases. The production of $CH_4$ in wetlands depends on carbon availability, soil redox potential, availability of $O_2$, pH, and temperature, and it is released from wetland soils by ebullition, convection, diffusion, and ventilation [128,129]. Thus, $CH_4$ emissions depend on the plant species present, and their abundance [130–132].

A key role in greenhouse gas emission is also played by different growth forms of plants. On one hand, Kosten et al. [133] find that free-floating plants may reduce methane emissions. On the other hand, some floating-leaved species contribute significantly to methane emission, including *N. odorata* [20], *N. nucifera* [19], and *N. luteum* L. [20], and helophytes such as *Juncus effusus* L. [134], *Spartina anglica* and *P. australis* [135,136], *P. mauritianus* Kunth [137], *T. latifolia* [137], *Typha* spp. [20,130], and *Cyperus papyrus* L. [137,138]. In areas colonized by *P. australis* and *S. lacustris,* methane is emitted exclusively by plant-mediated transport, with the highest emission rates at daytime, and emission peaks following sunrise [139].

The effects of aquatic plants on methane emission are complex [140]. Their dead and decaying remains provide the raw material for methane production, and their aerenchyma and gas canals facilitate the transport of methane to the atmosphere. At the same time, the aeration of their rhizospheres, and their methane-oxidizing microorganisms, decrease the amount of methane that escapes into the atmosphere [141]. In rice, two mutants with decreased root aerenchyma cause 70% less methane emission to the atmosphere than the wild type, while the oxygen transport is only reduced by 50% [142]. However, it is unlikely that such mutants will be a practical solution to the very high contribution of rice paddies to the greenhouse gas budget.

Wetland trees contribute to methane emission, while rainforest trees may function as sinks [143], except when flooded [144,145]. In *Phragmites* spp. [136], pressurized gas flow is important in emission; in the case of these species, the population of methane-producing microbes in the rhizosphere may be the most crucial factor. The relative importance of these factors is discussed by Laanbroek [146].

## 5. Conclusions

Ventilation in different groups of wetland plants that thrive in oxygen-deficient soils is related to aerenchyma as a ventilation path; however, the mechanisms may differ significantly. This diversity is a consequence of (1) phylogeny, as waterlogging presents an environmental filter that excludes certain species; (2) specific environmental conditions in plant/species habitat; (3) specific morphological adaptations at plant/species level (as is the case for water lily and lotus); (4) growth form (e.g., emergent, submerged, floating-leaved, rosette, woody plants) is related to plant medium (air, water, or both); and (5) metabolic processes (photosynthesis, respiration, photorespiration) that may act as a source or a sink of gases. These diverse structures and processes are related to ventilation support, photosynthesis, and respiration processes. Along with plant ventilation, they contribute to the oxygenation of otherwise anoxic wetland soils by radial oxygen loss from roots to the rhizosphere, and, thus, its detoxification, and also facilitate the release of greenhouse gases from these soils, which sets off a cascade of environmental consequences.

**Author Contributions:** Conceptualization, L.O.B. and A.G. writing—original draft preparation, L.O.B., B.A.M., M.G. and A.G., writing—review and editing, L.O.B., B.A.M., M.G. and A.G.; visualization, L.O.B., A.G. and B.A.M.; supervision, L.O.B., B.A.M. and A.G.; funding acquisition, A.G. and B.A.M. All authors have read and agreed to the published version of the manuscript.

**Funding:** This study was supported by the Slovenian Research Agency, through the Plant Biology program (P1-0212), and the U.S. Geological Survey Ecosystems Mission Area.

**Institutional Review Board Statement:** Not applicable.

**Informed Consent Statement:** Not applicable.

**Data Availability Statement:** Not applicable.

**Conflicts of Interest:** The authors declare no conflict of interest.

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
