# Peer review of "Ventilation Systems in Wetland Plant Species"

_diversity, doi:10.3390/d14070517_

Round 1

Reviewer 1 Report

The manuscript entitled Ventilation systems in wetland plant species [ID: diversity-1725434] describe the study of aerenchyma and ventilation mechanisms in different ecological groups of wetland plants and potential effects on the environment. In particular, information on morphological adaptation of various wetland plants was provided through various studies by classifying wetland plants based on their ecological characteristics.

The manuscript needs structural revision. Sections of 1.1-1.7 need to be categorized by contents and not at the same level. For example, (1.1~1.5), (1.6) and (1.7) appear to exhibit different characteristics in terms of content. In other words, it appears that rearrangement or modification of the structure is necessary overall (1.6→2, 1.7→3, 2→4). This will help the reader improve his or her understanding and readability of the content.

In addition, it is necessary to write a clearer research purpose in the introduction part and write the answer to the purpose of this thesis in more detail in the conclusion part.

Author Response

We thank you for valuable comments that helped us to improve the manuscript. Below you will find the responses to comments of reviewers. We took into account your corrections and recommendations.

The manuscript entitled Ventilation systems in wetland plant species [ID: diversity-1725434] describe the study of aerenchyma and ventilation mechanisms in different ecological groups of wetland plants and potential effects on the environment. In particular, information on morphological adaptation of various wetland plants was provided through various studies by classifying wetland plants based on their ecological characteristics.

The manuscript needs structural revision. Sections of 1.1-1.7 need to be categorized by contents and not at the same level. For example, (1.1~1.5), (1.6) and (1.7) appear to exhibit different characteristics in terms of content. In other words, it appears that rearrangement or modification of the structure is necessary overall (1.6→2, 1.7→3, 2→4). This will help the reader improve his or her understanding and readability of the content.

Response: Corrected as suggested.

In addition, it is necessary to write a clearer research purpose in the introduction part and write the answer to the purpose of this thesis in more detail in the conclusion part.

Response: We make the purpose clearer and conclusions  more firm.

Reviewer 2 Report

The ms. entitled: Ventilation systems in wetland plant species ms. by Björn et al. is a revision on the mechanisms and processes involved in aquatic plant air transport for photosynthesis and respiration organized for the different functional aquatic plant groups. As such it represents an interesting and useful compilation of information, easy to read, except for poor narrative in few sections, but standing as a basic compilation of information. In my opinion the work would benefit from adding additional information on the metabolic mechanisms that contribute to ventilation, this is a term that appears often in the text but it is not really defined. A second point that should improve the interest of the text would be adding information on whether the functional traits and mechanisms follow the phylogenia or by contrary they are common secondary adaptations, and providing examples of the two situations if it is the case. This is a topic that it is mentioned a bit but not profundly. It would also be nice to improve the objectives of the work and, finally, the conclusions should refer to the adaptative topic I just mention plus other conclusions omitted in the present paragraph.

Speciphic comments:

L. 45-50 changes in the CO2 and O2 in the aerenchima, where? 

L. 54-55 set at genus level, please add reference

L. 87-89 I suggest hypothesis driven objectives based on functional groups for example. Please explain the term ecological groups, I am not familiar to it.

L. 92-93 check narrative

L. 100 you mean 'is also true'

L. 100-102 check narrative

L. 185-189 be more informatibe other than historical report

L. I do not remember that you had explained laminal stomata before, please do so.

Author Response

Please see attached the author's updated responses.

Author Response

We thank you for valuable comments that helped us to improve the manuscript. Below you will find the responses to comments of reviewers. We took into account your corrections and recommendations.

The authors present a review of ventialtion systems in different types of wetland plants. I found the paper very itneresting but the english is often rather terrible. The disposition of the texts is acceptable, the pghrasing is however often poorly written and unclear. A few phrases are quite naive. There is some minor erre in the nomenclature.

Below I tried to indicatesome way to fix these problems, I think that with some (much) work the paper can be greatly improved and made more acceptable.

Abstract

15 aerenchyma cells with varied structur es:-> various arenchymatic structures

Response: Corrected

Introduction

The introduction does not intorudces well the argument; probably you shold put the subpargagraph 1.6 here and define the terms and concepts that are ncessary for understanding the discussion of differente types of wetland plants; also aa definition of wetland plants and the main subdivisions of this groups could be usefully added.

Response: We added the paragraph about different types of wetland plants and their specifics.

37 Connecting them to the atmosphere : please delete it is redundant

Response: Deleted.

49 delete cells

Response: Deleted.

49-51 the sentence is unclear

Response: Corrected.

44 what do you mean by set at the genus level?

Response: This part of the sentence was deleted, because it is redundant, the information is given in the next sentence.

68 ratio-> amount

Response: Corrected.

85 via aerenchyma at the end of the sentece

Response: Corrected.

76 this critical oxygen -> this critically importatn oxygen

Response: Corrected.

102 Lobelia dortmannana etc lacks a verb

Response: Corrected.

118 limited light and carbon dioxyde supply

Response: Added.

119 please add a reference

Response: Added.

120 presente day… i a repetition of what said above

Response: Omitted.

126 again a redudance

Response: This sentence refers to modern species.

127-128 this important concept is introduce here but whas mentionead earlier; probably it should be put more at the beginnin

Response: This concept refers to the modern Isoetids.

129 which are present in various … is again a repetition

Response: This sentence refers to Figure 2.

131 Isoetes can be either lacustrin or amphibous; please delete lacustrine

Response: Omitted.

dele133 you did not define isoetedi

Response: Definition added.

 etc.

143-144 what do you mean by aeretion of the sediment= probably that they can survive obnly uif sufficient oxyegn reaches the ground bu this is in contradition or at least unrelatedf with the following sentence

Response: Written clearer.

151-153 this was already mentioned before buti s better placed here

Response: This refers specificaly to Nuphar advena.

158 please delete the rethoric question

Response: Deleted.

162-163 poorly written

Response: corrected.

Response: Written clearer.

172-163 poorly written and the italic is unncessary (it seems a stress but that is better expressed in some other way)

Response: Corrected, written clearer.

179 uinterecellulars-> intercellular spaces

Response: Corrected.

187 Nuphar luteum not lutea

Response: Corrected everywhere.

186-189 and what are the main results?

Response: Added.

191 we reproruce_>  simply quote Tab…

Response: Corrected.

Figure 4 very interesting; the text is not equally clear

Response: Written clearer.

217 please add the authors to Polygonum amphibium; most authors recombine this species in Persicaria

Response: Added.

228 transection-> section

Response: Corrected.

226 above ground organs of helophytes are always in air, it is the definiton of helophytes; probably the author mean that they can be occasionally flooded

Response: Yes.

228-229 terms rather unprecise

Response: We deleted this sentence.

234 Equisetum spp. Without authro

Response: Corrected.

253 Phrtagmites spp without author

Response: Corrected.

256 delete so-called

Response: Deleted.

287 I think the taxonomic information is unnecessary or otherwise shpuld be provided for alla species; if necessary you can simply write that cattails are phylogenetically related to reed

Response: Corrected.

309-310 literature cited by Raskin and kende is quoteterrible; please indicate references even if taken from Raskin and kende

Response: These references are not accessible, so we changed text accordingly and added two additional references.

340 waht do ypu mean by even the roots?

Response: Including.

355 and following probably you can put the mangroves at the beginning and the other species at the end. I understand the reasons for the order that you have followed buti t is not very clear; consider also renaming the tietle with, simply, mangorve forests

Response: The title corrected. We let the order as it was, because we took mangrove forest as a special wetland.

357 cite Fi,.g 9 and delate lines 364

Response: Somewhat corrected.

411-412 put the equation in its own line and write it with some programs for wirting mathemtaical formulas

Response: Corrected.

415-4519 poorly written

Response: Somewhat corrected.

Paragraph 1.6 this is better placed at the beginning

Paragraph 1.6 this is a huge subject, and perhaps you should emphasize only the role of vebntilation in greenhous gases reduction. POtherwhise provide a deepr end more estensive discussion of the subject

Response: We are aware, we just wanted to mention this aspect (and not go into details), because it is a by-effect of ventilation system. We re-numbered the paragraphs, now so it is special chapter.

441-442 I agree but the sentence is rather redundant. Please delete

Response: Corrected.

457 Typha without author

Response: Corrected.

Round 2

Reviewer 3 Report

the paper after the revisions has greatly improved.